# TChange: A Hybrid Transformer-CNN Change Detection Network

Yupeng Deng [1,2,3], Yu Meng [1,*], Jingbo Chen [1], Anzhi Yue [1], Diyou Liu [1] and Jing Chen [1]

1   Aerospace Information Research Institute, Chinese Academy of Sciences, Beijing 100094, China
2   University of Chinese Academy of Sciences, Beijing 100049, China
3   Electronic, Electrical and Communication Engineering, University of Chinese Academy of Sciences, Beijing 100190, China
*   Correspondence: mengyu@aircas.ac.cn; Tel.: +86-010-64868230

**Abstract:** Change detection is employed to identify regions of change between two different time phases. Presently, the CNN-based change detection algorithm is the mainstream direction of change detection. However, there are two challenges in current change detection methods: (1) the intrascale problem: CNN-based change detection algorithms, due to the local receptive field limitation, can only fuse pairwise characteristics in a local range within a single scale, causing incomplete detection of large-scale targets. (2) The interscale problem: Current algorithms generally fuse layer by layer for interscale communication, with one-way flow of information and long propagation links, which are prone to information loss, making it difficult to take into account both large targets and small targets. To address the above issues, a hybrid transformer–CNN change detection network (TChange) for very-high-spatial-resolution (VHR) remote sensing images is proposed. (1) Change multihead self-attention (Change MSA) is built for global intrascale information exchange of spatial features and channel characteristics. (2) An interscale transformer module (ISTM) is proposed to perform direct interscale information exchange. To address the problem that the transformer tends to lose high-frequency features, the use of deep edge supervision is proposed to replace the commonly utilized depth supervision. TChange achieves state-of-the-art scores on the WUH-CD and LEVIR-CD open-source datasets. Furthermore, to validate the effectiveness of Change MSA and the ISTM proposed by TChange, we construct a change detection dataset, TZ-CD, that covers an area of 900 km$^2$ and contains numerous large targets and weak change targets.

**Keywords:** deep learning; change detection; convolutional neural network; multihead attention; transformer

## 1. Introduction

Change detection (CD) captures the spatial changes between two multitemporal satellite images due to manmade or natural phenomena [1]. Pixels in the same region but acquired at different times are usually classified as changed or unchanged by comparing the coregistered images [2]. The development of the CD algorithm is mainly divided into two aspects. First, many researchers analyze the imaging mechanism and feature design. Second, the resolution of satellite data is constantly improving, from a spatial resolution of 79 m (Landsat-1 satellite) to today's massive submeter satellite data, which provides experimental scenarios and further development directions for research on change detection algorithms.

Traditional CD algorithms achieve better detection results on low-resolution data by manually and elaborately designing features and adjusting super parameters. At the early stage of development, researchers are directly comparing and identifying changes in Landsat images by image differentiation, imaging and regression analysis [3–5]. Based on the method of feature space mapping, principal component analysis (PCA) [6] is utilized to compress features to obtain the main features, and then changed regions are obtained by the change vector analysis (CVA) [7] algorithm. However, the above methods have high data requirements and require the same sensor and data with consistent radiation

characteristics as much as possible [8]. With an increase in the number of satellites, the use of classification-based CD methods has increased [9–11]. This method classifies first and then detects changes, avoiding the problem of false changes caused by inconsistent radiation. Therefore, the change problem is transformed into how to classify the surface features with high quality.

With the development of satellite imaging and the launch of numerous land satellites, high-resolution images have become increasingly accessible. In China, business departments use images with resolutions from 0.5 to 2 m for national land cover detection and change monitoring. In the SpaceNet 7 challenge, the high-resolution data of SPOT are used for competition, which has higher requirements for extracting change target technology. The traditional CD algorithm based on manually designed features and many super parameters has poor performance because it is difficult to model semantics on high-resolution images [12–14]. Thus, this algorithm is gradually replaced by a deep neural network that is based on numerous data and samples.

Presently, algorithms based on convolutional neural networks (CNNs) have shown excellent performance in a variety of change detection tasks. These networks extract multilayer pyramid features via encoders and fuse change features via decoders [15–26]. Based on the encoding and decoding structure, numerous researchers have made different innovative works according to the characteristics of different change detection tasks. Chen et al. designed the encoder by residual connection and pretrained it using a large change detection dataset [22]. Daudt et al. decoupled the dual-phase features and introduced the twin encoder for independent feature extraction [15]. Some scholars have investigated the fusion method of dual characteristics after the twin network has extracted features and proposed more efficient data fusion methods that integrate prior knowledge of change detection [15,19,21,22,27,28]. In addition, multiscale information exchange channels can be added to the decoder, and change decoding can be better handled by adding depth supervision [21]. In view of the lack of change samples, we proposed using segmentation information to guide encoder learning [29,30] and set specific data enhancement schemes for change detection tasks to greatly improve the utilization efficiency of sample information [31].

However, the algorithm encounters some problems based on CNNs. (1) The receptive field is limited. CNNs expand the receptive field via downsampling, which easily causes information loss. (2) It is difficult to exchange information across scales. The multiscale features of CNNs are usually sampled layer by layer [32] or directly concatenate [33]. The information across scales and channels is combined, which makes it difficult to obtain information and improve the accuracy. The final performance on the change detection task is incomplete detection of large targets and easy loss of small targets.

Recently, the appearance of a transformer [34] provided a new feature extraction method. Using a self-attention mechanism, each pixel can obtain global information. With the application of a transformer to image segmentation [35–38] and detection [39] tasks, the accuracy has been greatly improved. In the field of change detection, BIT [40] uses a convolutional neural network to extract features and then constructs the encoder-decoder of a transformer to obtain the features at different times. Although a transformer is utilized, due to the inappropriate method of the transformer and CNN hybrid, its accuracy is inferior to that of a CNN. In subsequent research, researchers used the transformer to fuse twin network features under the overall architecture of a CNN network [21,23] and achieved better accuracy than a pure CNN. DMATNnet [41] proposed a fuse the fine and coarse features with dual-feature mixed attention base on transformer. It can not only extract more specific regions of interest, but also overcome the misjudgment caused by oversampling, and synchronize feature extraction and target information integration. SwinSUNet [42] Unet builds encoders and decoders based on SWIN [36] to achieve leading results across multiple data sets. Transformers can not only provide global feature modeling to obtain remote semantic information but also build information exchange channels between two features [23]. However, the original MSA disregards the biased induction characteristics of

CNNs and the computational complexity of MSA when modeling. Network computing is slow, and convergence is difficult when the dataset is small. Although the transformer can provide a higher accuracy, there is a large demand for data, and the original multihead self-attention (MSA) [43] involves extensive computation.

The development of the current change detection task is inseparable from the support of many open datasets. The public dataset provides an open benchmark for algorithms so that different algorithms can be compared on the same basis. WUH-CD [44] and LEVIR-CD [45] provide extremely high-resolution building changes, and SECOND [46] provides multicategory change labels. However, in the actual change detection scene, there are numerous large targets, which have regions with weakened changes and regions with obvious changes. Usually, people need to infer weak regions by regions with obvious changes during detection. However, this scenario does not exist in the current public dataset. Therefore, this paper constructs a change detection dataset with a large map, which can complement the lack of change scenarios in the current public dataset.

In this paper, to solve the problem that it is difficult to completely detect large targets and lose small targets in the current CNN, a Change MSA module is designed on the intrascale by using the global modeling capability of a transformer. For the problem that the original MSA involves extensive computation, a block MSA named S-MSA is utilized. To solve the problem of change data fusion, the original MSA spatial dimension construction tokens are replaced by the channel construction tokens, thus realizing the direct global feature exchange of channel dimensions. To solve the problem of the large data demand of the transformer, MSA and a feature fusion module (FFM) are combined to enhance the local feature modeling of the transformer by using the biased induction feature of convolution and to greatly reduce the computational requirements. Through the multiscale characteristics of blocks, the module can efficiently model the characteristics of different scales while maintaining low computational complexity. The characteristics of the transformer and the twin characteristics extracted by the encoder are merged layer by layer under the CNN decoder so that the network can simultaneously extract large and small targets while maintaining the high precision of the original CNN and can use the regions with obvious changes to enhance the characteristics of the regions with weak changes.

The contributions of this work are summarized as follows:

- We propose a hybrid transformer–CNN change detection network named TChange. Under the condition of maintaining a low computational cost, the network can globally and efficiently model the features within the scale and provide a direct information exchange channel for features across scales.
- A novel MSA module named Change MSA is proposed to acquire global feature and pairwise feature information within scales. In addition, a new feature fusion method, which conducts MSA companies in the channel dimension rather than the spatial dimension by channel crossing, is proposed for change detection tasks. The offset induction of CNNs is used to enhance the local modeling ability of MSA.
- An interscale transformer module (ISTM) is proposed to build a multiscale feature exchange channel.
- A new remote sensing change detection dataset named TZ-CD is constructed by taking into account changed regions with various areas, which compensates for the lack of scenarios in the current change detection public dataset.

## 2. Materials and Methods

### 2.1. Overview

TChange uses a transformer and CNN to build a model that can provide high-precision pixel-level change area detection. The overall structure of TChange is shown in Figure 1. First, the network extracts multilevel pyramid features via multiple feature extractors. Second, to solve the problem of multiscale feature fusion, the common approach is to use a sampled CNN to fuse layer by layer so that shallow features can obtain the information of deep features. However, the flow of information is one-way, some small targets in

the deep features easily lose information, and deep features have the problem of a local receptive field. Therefore, this paper proposes an interscale information fusion module based on a transformer. While preserving the layer-by-layer fusion decoding method, the information of multiscale features is exchanged by the long-distance modeling capability of the transformer. Last, because it is easy to lose high-frequency features in the transformer and the decoding module is more complex, to enhance the learning ability of the model, in-depth supervision is utilized to learn the edges of various sizes to ensure that the high-precision edges are obtained while the changing target is detected.

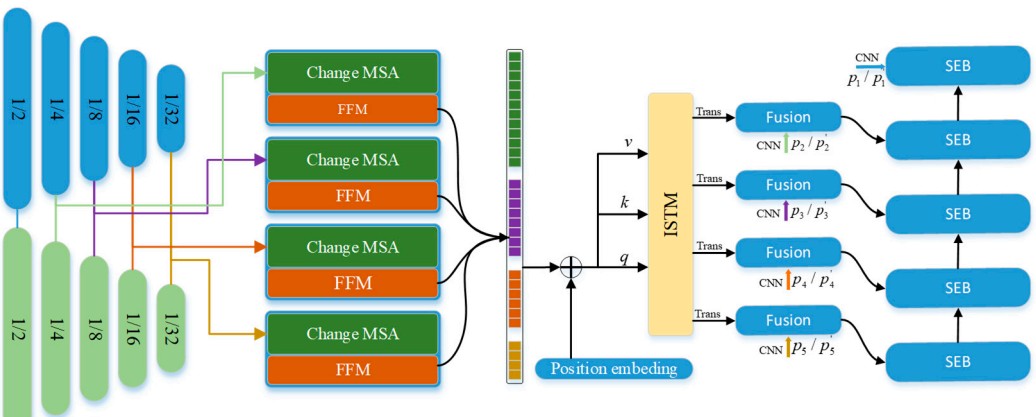

**Figure 1.** Overall structure of TChange, which consists of an encoder, Change MSA, an interscale transformer module (ISTM) and a CNN decoder. FFM indicates the feature fusion module. SEB denotes the segmentation module.

### 2.2. Encoder

TChange is aimed at high-resolution remote sensing images. The structure of TChange includes a CNN–transformer hybrid decoder, so the computation is heavy. Therefore, to balance efficiency and accuracy, the lightweight and efficient b1is selected as the feature extractor to extract the multilayer pyramid features. The input of TChange is remote sensing image data in different time phases: $X_{post}, X_{pre} \in \mathbb{R}^{H \times W \times 3}$. Multiscale features are aligned by twin encoders that share weight, and two groups of multiscale features $\{p_1, p_2, p_3, p_4, p_5\}$ and $\{p'_1, p'_2, p'_3, p'_4, p'_5\}$, whose spatial scales are $\left\{ \frac{h}{2} \times \frac{w}{2}, \frac{h}{4} \times \frac{w}{4}, \frac{h}{8} \times \frac{w}{8}, \frac{h}{16} \times \frac{w}{16}, \frac{h}{32} \times \frac{w}{32} \right\}$, are output. TChange uses a universal encoder, which can directly use the ImageNet pretrained model to obtain higher accuracy and faster training speed.

### 2.3. ChangMSA

In high-resolution remote sensing images, there are many large change targets, and there are weak change features in each large change target. When people interpret visually, they gradually index the areas with weak change according to the areas with obvious change characteristics. Previously, some scholars applied the UNet network structure to fuse features of different scales layer by layer to identify changes in weak areas by using remote feature information. The rise of transformers provides a more efficient way to obtain remote semantic information without downsampling. Therefore, this paper proposes an MSA-based approach to discover changes and obtain remote semantic information.

To solve the problem of long-distance semantic acquisition and feature pair change discovery, this paper designs an MSA module for change detection tasks based on MSA, named Change MSA, which consists of three parts: (1) C-MSA for interchannel awareness, (2) S-MSA for spatial long-distance semantic acquisition, and (3) FFM for convolutional biased induction to enhance local feature learning.

Before introducing our specific implementation, first, we describe the basic paradigm of MSA in image segmentation. The input is $x \in \mathbb{R}^{H \times W \times C}$, which is reshaped to tokens $HW \times C$, where the dimension of each token is $C$. The tokens are linearly projected into $query\ Q \in \mathbb{R}^{HW \times C}$, $key\ K \in \mathbb{R}^{HW \times C}$, and $value\ V \in \mathbb{R}^{HW \times C}$. Second, the input is sent to multiple heads for self-attention calculation, and each single head is calculated as follows:

$$A(Q, K, V) = softmax\left(\frac{Q \cdot K^T}{\sqrt{d}}\right) \cdot V \tag{1}$$

$K^T$ is the transposition of K, and $\sqrt{d}$ is the normalization factor to avoid large values after the dot-product operation. If it is a multihead self-attention network, the number of heads is N, and the dimensions of $Q$, $K$ and $V$ are divided into N parts and then output to the channel dimension $Concate$. The computational cost of the original MSA is $O(H^2 W^2 C)$.

The original MSA is intended to obtain long-distance semantic information. However, the change detection task requires not only spatial long-distance information but also the information between two change features. Therefore, the Change MSA proposed in this paper includes two parts, channel awareness and spatial awareness, which are detailed in the following section.

### 2.3.1. C-MSA

Change detection uses twin encoders to obtain multilevel pyramid features. In most papers, the fusion method of features is concate and the change area is simply and roughly obtained by the learning ability of deep learning. From the perspective of prior knowledge, this paper designs a C-MSA based on the original MSA network.

As shown in Figure 2, the input data are $p_n \in \mathbb{R}^{H \times W \times C}$, $p'_n \in \mathbb{R}^{H \times W \times C}$. We suggest that different channels represent the detection results of the same detector. With twin encoders sharing weight, the same level is the result of the same detector. Therefore, all channels are crossed to form feature $x_{in} \in \mathbb{R}^{H \times W \times 2C}$.

$$X_{in} = Concate(p_n, p'_n),\ n \in [2, 3, 4, 5] \tag{2}$$

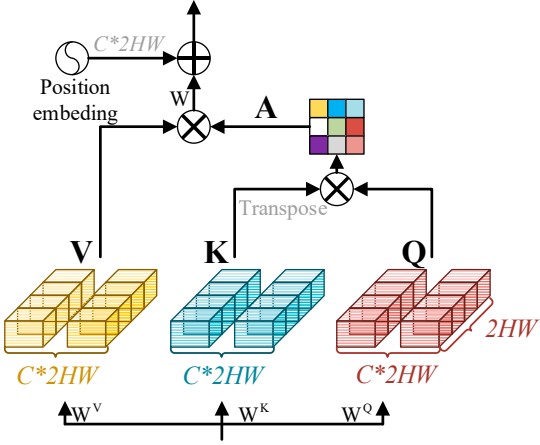

**Figure 2.** Details of the C-MSA structure. Each token is formed by flattening the spatial dimension.

Two channels are combined into a group to build $C$ tokens: $\left[\{p_1, p'_1\}, \ldots, \{p_c, p'_c\}\right]$. Note that the subscripts at this time represent different token indices. Since C-MSA uses the paired spatial regions as a token, each token is reshaped from $H * W * 2$ to $2 * H * W$, and its feature depth is $2HW$. Compared with the token of the original MSA, as shown in Figure 3, the original MSA uses the spatial feature as the token, and there are HW tokens (Figure 3a). This paper uses the dual feature channel as the token, so there are $C$ tokens (Figure 3c).

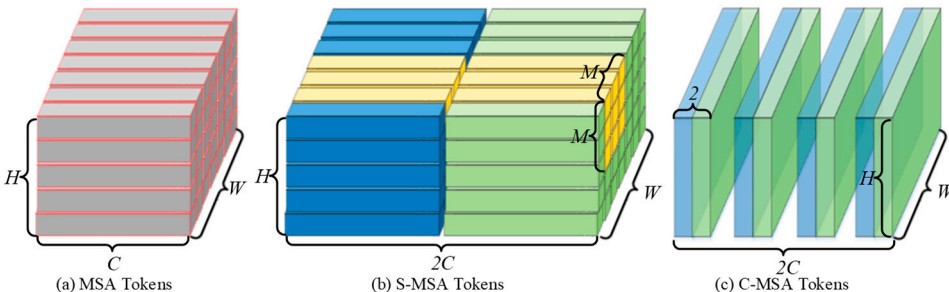

**Figure 3.** (**a**) Original MSA tokens; (**b**) S-MSA tokens, where multiple yellow areas comprise a token; (**c**) C-MSA tokens, channel crossing, and building tokens along the channel dimension.

The dimension of each token is 2 HW, and then tokens are linearly projected into *query* $Q \in \mathbb{R}^{HW \times C}$, *key* $K \in \mathbb{R}^{HW \times C}$, *value* $V \in \mathbb{R}^{HW \times C}$.

$$Q = XW^Q, \ K = XW^K, V = XW^V \tag{3}$$

Next, we cut *Q*, *K*, and *V* into *N* pieces along the dimension direction and input N heads, with $Q = [Q_1, \ldots, Q_N], K = [K_1, \ldots, K_N], V = [V_1, \ldots, V_N]$. The dimension of each head is $C/N$. Thus, the calculation of the jth head is expressed as follows:

$$A_j = softmax\left(\frac{K_j^T \cdot Q_j}{\sqrt{d}}\right), H_j = V_J \cdot A_j, \ j \in (1, N) \tag{4}$$

$$C - MSA(X_{in}) = concate(H_1, \ldots, H_N)W + f_p(V) \tag{5}$$

$K_j^T$ is the transposition of $K_j$, $\sqrt{d}$ is a normalization factor to avoid large values after the dot-product operation, and $W \in \mathbb{R}^{2C \times 2C}$, and $f_p(\cdot)$ is a function for encoding the channel position. The position encoding of the original MSA is used to encode spatial information, while the C-MSA flattens the channel into a token, so its position encoding is utilized to encode channel position information. Therefore, the final output of C-MSA is $X_{COUT} \in \mathbb{R}^{H \times W \times 2C}$.

Compared with the computational complexity $O(H^2W^2C)$ of the original MSA, the computational complexity of C-MSA is $O(2HWC^2)$. It can be seen that the calculation amount of C-MSA linearly increases with the spatial resolution, while the original MSA exponentially increases when carrying out large image detection of remote sensing images.

### 2.3.2. S-MSA

To ensure the edge texture of the target and realize the detection of small targets, spatial MSA (S-MSA) is used in the 1/4 downsample features. The original MSA has the problem of high computational complexity. Therefore, this paper uses SWIN [36] for reference and adopts the sliding window method. The original MSA is based on a single pixel, while S-MSA is based on the nonoverlapping window of $M^2$, which reduces the number of tokens from $HW$ to $\frac{HW}{M^2}$. Tokens are shown in Figure 3b, which aggregates the original yellow area into a token pixel by pixel. The subsequent S-MSA is calculated in the same way as the original MSA.

The S-MSA computational cost is $O(2M^2HWC)$. S-MSA also linearly increases the amount of computation with the spatial resolution. Thus, TChang can use MSA to obtain long-distance spatial information at a higher resolution.

### 2.3.3. FFM

The feature fusion module (FFM) is aimed at using offset induction of the convolutional network to obtain local features, which is applied to strengthen the global feature modeling of MSA. The process is described as follows: first, restore the output feature of MSA to the spatial dimension X. Second, input a $3 \times 3$ convolution to learn inductive biases, and

then input the feature to the Layernrom and GELU activation layers. Last, obtain the final output via a $3 \times 3$ convolution layer.

$$FFM(X) = Conv_{3*3}(GELU(LN(Conv_{3*3}(X)))) \tag{6}$$

### 2.3.4. Change MSA Summary

The structure of Change MSA is shown in Figure 4. The paired feature pairs $p_n / p'_n$, $n \in [2, 3, 4, 5]$ are extracted using four groups of encoders. The features p and q in the channel dimension are interleaved to obtain $X_{in}$.

$$X_{cout} = C - MSA(LN(X_{in})) + X_{in} \tag{7}$$

$$X_{ffm\_0} = \text{FFM}(\text{LN}(X_{cout})) + X_{cout} \tag{8}$$

$$CM_n = \text{FFM}\left(\text{LN}\left(\text{S} - \text{MSA}\left(X_{ffm_0} + P_m\right) + \left(X_{ffm_0} + P_m\right)\right)\right) n \in [2, 3, 4, 5] \tag{9}$$

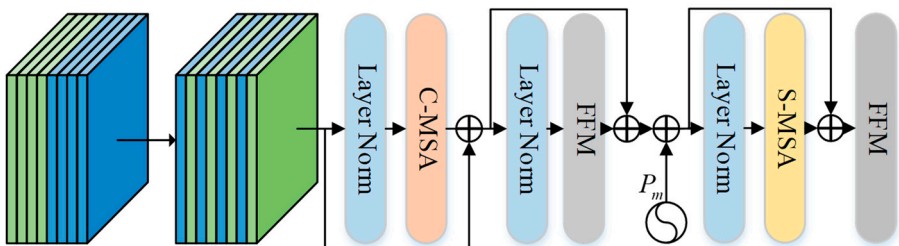

**Figure 4.** Change MSA overall structure.

In the above formula, LN represents the layer norm, and $P_m$ denotes the position encodings, which are composed of convolution $1 \times 1$. First, $X_{in}$ is input via layer normalization, C-MSA is input for channel feature coding, and then the output features and $X_{in}$ are added to obtain $X_{cout}$. X is input into the FFM to strengthen local features, S-MAS is input, and the final output $CM_n$ $n \in [2, 3, 4, 5]$ is obtained via the FFM module. Its spatial scale is $\left\{ \frac{h}{2} \times \frac{w}{2} \times 128, \frac{h}{4} \times \frac{w}{4} \times 128, \frac{h}{8} \times \frac{w}{8} \times 128, \frac{h}{16} \times \frac{w}{16} \times 128, \frac{h}{32} \times \frac{w}{32} \times 128 \right\}$. It should be noted that in order to balance precision and computation, TChange adds Change MSA at the beginning of 1/4 scale.

### 2.4. The Inter-Scale Transformer Module

In change detection research, the transmission and fusion of feature information at different scales greatly improve the detection accuracy. This paper proposes a scale feature exchange mechanism based on a transformer, which is aimed at directly exchanging information about features at different scales using the characteristics of the transformer. The design of this module has two characteristics: (1) Building a characteristic communication channel between two scales and (2) less computation.

Different from the original MSA, the proposed ISTM input is a multiscale feature. To exchange information on different scale features, we flatten the feature maps on different scales and then simultaneously concatenate them. Through the self-attention mechanism of the transformer, each spatial feature can obtain global feature information. To further optimize the calculation amount, we divide different scale features into blocks with the same spatial dimension, as shown in Figure 5. The same block area with different scales corresponds to the same area in the original data, so that the calculation amount and spatial dimension can be linearly related while realizing channel exchange.

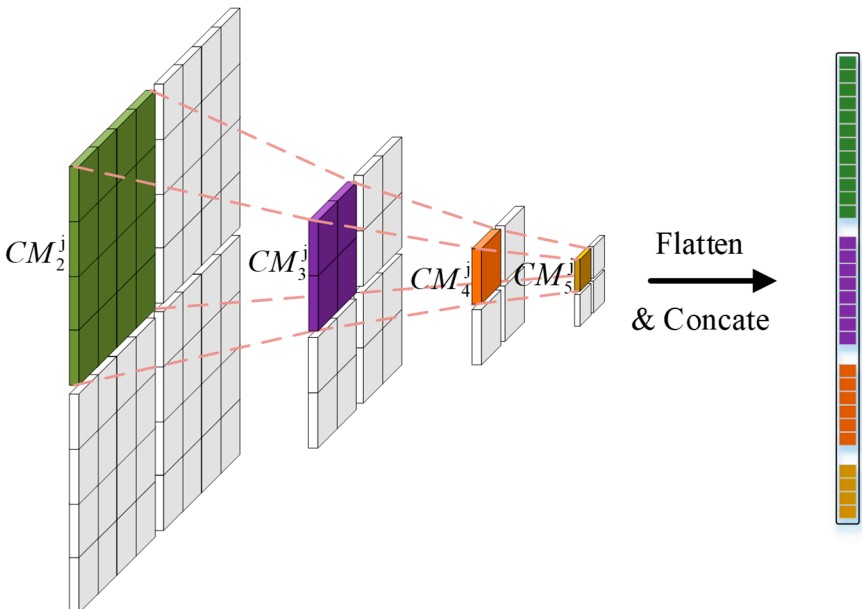

**Figure 5.** Different scale features into blocks. Different colors represent different scale features.

The specific implementation is described as follows:

The input of this module is a multiscale feature map, $CM_n$ $n \in [2, 3, 4, 5]$, which first divides the features of different scales into N regions. The same regions of different scales have different sizes, but they are features of different levels of the same region of the input data. The features are flattened and then spliced to obtain the input $S_j$, $j \in \{1, N\}$.

$$S_j = Concate\left(flatten(CM_2^j), flatten(CM_3^j), flatten(CM_4^j), flatten(CM_5^j)\right) \quad (10)$$

where $flatten(\cdot)$ is a function that expands the spatial dimension of $CM_n^j$ and *Concate* is used to concatenate all flattened features into one feature.

In Change MSA, the space dimension and channel dimension are fully transmitted by self-attention, so the ISTM is mainly aimed at providing an information exchange channel between two features of different sizes. To reduce the calculation, features have been processed in blocks. The input token represents features of different scales in the same area at the pixel level. We use the original MSA to exchange features; it is expressed as follows:

$$S'_j = FFM\left(LN\left(MSA\left(LN(S_j)\right) + S_j\right)\right) + \left(MSA\left(LN(S_j)\right) + S_j\right) \quad (11)$$

where $LN$ denotes the layer normalization and $S'_j$ is reshaped to the original spatial dimension to obtain $\hat{CM}_n^j$ $n \in [2, 3, 4, 5]$. N regions are reassembled to obtain the output of the ISTM as $\hat{CM}_n$ $n \in [2, 3, 4, 5]$.

### 2.5. The CNN Decoder

The decoder consists of two parts: the fusion and segmentation block (SEB), which provides fusion transformer and CNN features, and then fusion decoding layer by layer via the SEB.

Fusion has 3 inputs, as shown in Figure 6a; the input of the CNN is $p_n/p'_n$, $n \in [1, 2, 3, 4, 5]$, and the input of the transformer is $\hat{CM}_n$ $n \in [2, 3, 4, 5]$. Compared with common feature fusion operations, such as *Concate*, this paper adds a priori knowledge of changes to the change module while maintaining no loss of features so that a faster training speed and higher accuracy can be obtained. The CNN input is described as follows:

$$C_n = Concate\left(\left|P_n - P'_n\right|, P_n + P'_n\right) n \in [1, 2, 3, 4, 5] \quad (12)$$

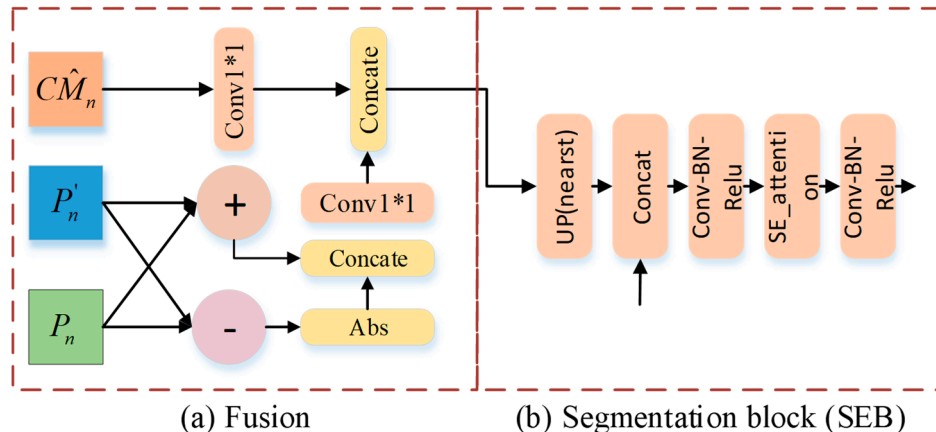

(a) Fusion      (b) Segmentation block (SEB)

**Figure 6.** (**a**) Fusion, in which features from the CNN and transformer are fused, and (**b**) segmentation block.

$Concate(\cdot)$ indicates that the features are concatenated in the channel dimension. When $n = 1$, there is no input from the transformer, so the $C_1$ results will be directly input into the SEB module. The overall fusion module is expressed as

$$F_n = \begin{cases} C_n & n = 1 \\ Concate(Conv_{1*1}(C\hat{M}_n), Conv_{1*1}(C_n)) & n \in [2, 3, 4, 5] \end{cases} \tag{13}$$

$Conv_{1*1}(\cdot)$ represents the convolution operation with a convolution kernel of 1 * 1. Features from the transformer and CNN are fused and complemented by fusion and input into the SEB.

**Segmentation block**: $F_n$ is set as the input to the upper sampling layer, and the spatial dimension is expanded from $H \times W$ to $2H \times 2W$. $F_n$ is fused with the features of the next scale and then combined with a set of $Conv - bn - relu$ operations. After $relu$, we use se attention [47] for spatial dimension feature fusion, which can better fuse the CNN and transformer features. The output $S_n$ $n \in [1, 2, 3, 4, 5]$ funded by the SEB module is obtained by a group of $Conv - bn - relu$. In special cases, when $n = 5$, because it is in the highest dimension, there is no fusion with the upper layer features, so the $Concate$ module is canceled, and the upsampling results are directly input to $Conv - bn - relu$.

### 2.6. Output and Deep Edge Supervision

To obtain $F_n$, $n \in \{1, 5\}$ from the decoder, in the TChange structure, we design two output modules: the segmentation head and edge head.

**Segmentation Head:** The input is the result $F_1$ of the last layer decoder. After convolving by Conv 3 * 3, the input is upsampled and activated via sigmoid to obtain the 0–1 probability map of the changed region, which is described as follows:

$$Pre_1 = Sigmoid(Bilinear(Conv_{3*3}(F_1))) \tag{14}$$

$Bilinear(\cdot)$ is a function for sampling twice the linear interpolation.

**Edge Head:** Because the decoder is heavy and the transformer is prone to losing high-frequency information, we adopt the method of deep supervision for each layer. Different from the commonly employed depth supervision, we replace pixel-level supervision with edge supervision to solve the problem that the transformer is prone to losing high-frequency information. Unlike the segmentation head, the edge head needs to actively guide high-frequency information. Therefore, we simultaneously perform average pooling and max pooling operations on $F_n$ $n \in [2, 3, 4, 5]$. Average pooling can be considered a balanced filter to obtain the low-frequency information of the feature map, while max pooling obtains the high-frequency information. The low-frequency and high-frequency information are

decoupled via pooling and then via a 3 * 3 convolution layer. The 0–1 probability map output is obtained using sigmoid.

$$Pre_n = Sigmoid(Conv_{3*3}([AvgPool(F_n), MaxPool(F_n)])) \; n \in [2, 3, 4, 5] \tag{15}$$

Note that the edge head only decouples high-frequency and low-frequency information to make it easier for the edge to learn.

In view of the small proportion of foreground categories in the prediction results, the Dice function is added on the basis of the commonly employed binary cross-entropy to optimize the foreground objectives. The weight of the two loss functions is 0.5:0.5.

## 3. Datasets

There are some publicly available datasets, but test scenarios where large-scale targets and small targets coexist, such as long-distance road changes, are lacking. Therefore, to address this particular lack of test scenarios, a new change detection dataset, TZ-CD, is proposed in this paper, and TChange is experimentally validated in TZ-CD. To demonstrate the generality of the algorithm in this paper, the LEVIR-CD [45] and WUH-CD [44] public benchmark datasets and the current state-of-the-art change detection algorithms are selected for comparison.

### 3.1. The Pubilc Dataset

The WUH-CD dataset is employed for the high-resolution building change detection task with a spatial resolution of 0.3 m. The pretemporal data were acquired in 2012, the post-temporal data were acquired in 2016, and the data size is 32,507 × 15,345, covering an area of 20.5 km$^2$. Because of the extremely high data resolution, the pixel size of its change target is large, as shown in Figure 7b, and its coverage area in the buildings is relatively low, which poses a higher challenge for the algorithm's adaptability in large targets.

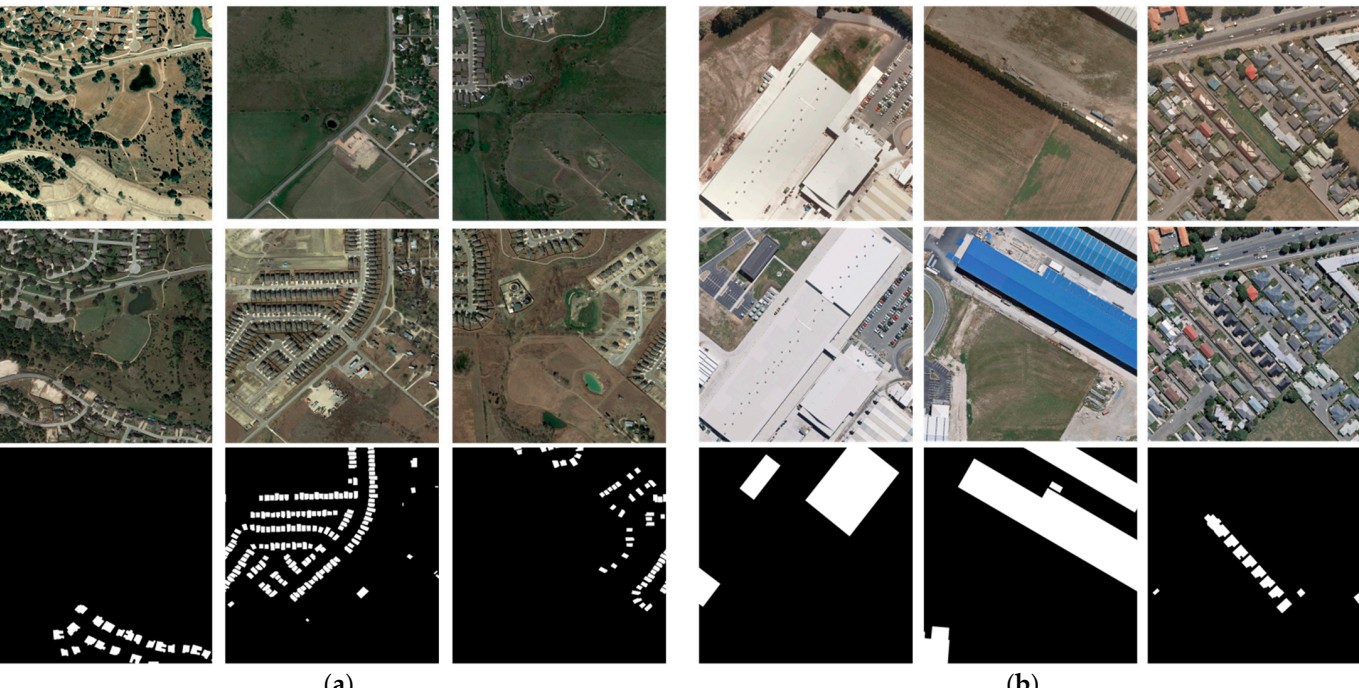

(**a**)　　　　　　　　　　　　　　　　　　　　　(**b**)

**Figure 7.** Front post-image, pre-image and ground truth from top to bottom. (**a**) LEVIR-CD dataset. (**b**) WUH-CD dataset.

The LEVIR-CD dataset is selected for the high-resolution building change detection task with a spatial resolution of 0.5 m and contains 637 pairs of data with a size of 1024 × 1024, of which 445 pairs are used for training, 64 pairs are utilized for validation, and 128 pairs are employed for testing. Some data examples are shown in Figure 7a, where the building targets are small and relatively dense.

### 3.2. The TZ-CD Dataset

The current commonly employed change detection dataset has a single data scale and scene. Thus, this paper proposes a new set of change detection data with a resolution of 1 m, covering an area of 900 km$^2$ and a size of 31,307 × 40,620, whose coverage area comprises the largest change detection dataset.

The dataset labeling includes categories for building change and push-fill change, and its precise labeling, with the presence of many small targets and large targets and containing areas of stronger change and areas of weaker change in the same target, presents new challenges to current change detection algorithms.

We divided this dataset into 3 regions, as shown in Figure 8, where the yellow region is used for testing, the red region is used for validation, and the other regions are used for training.

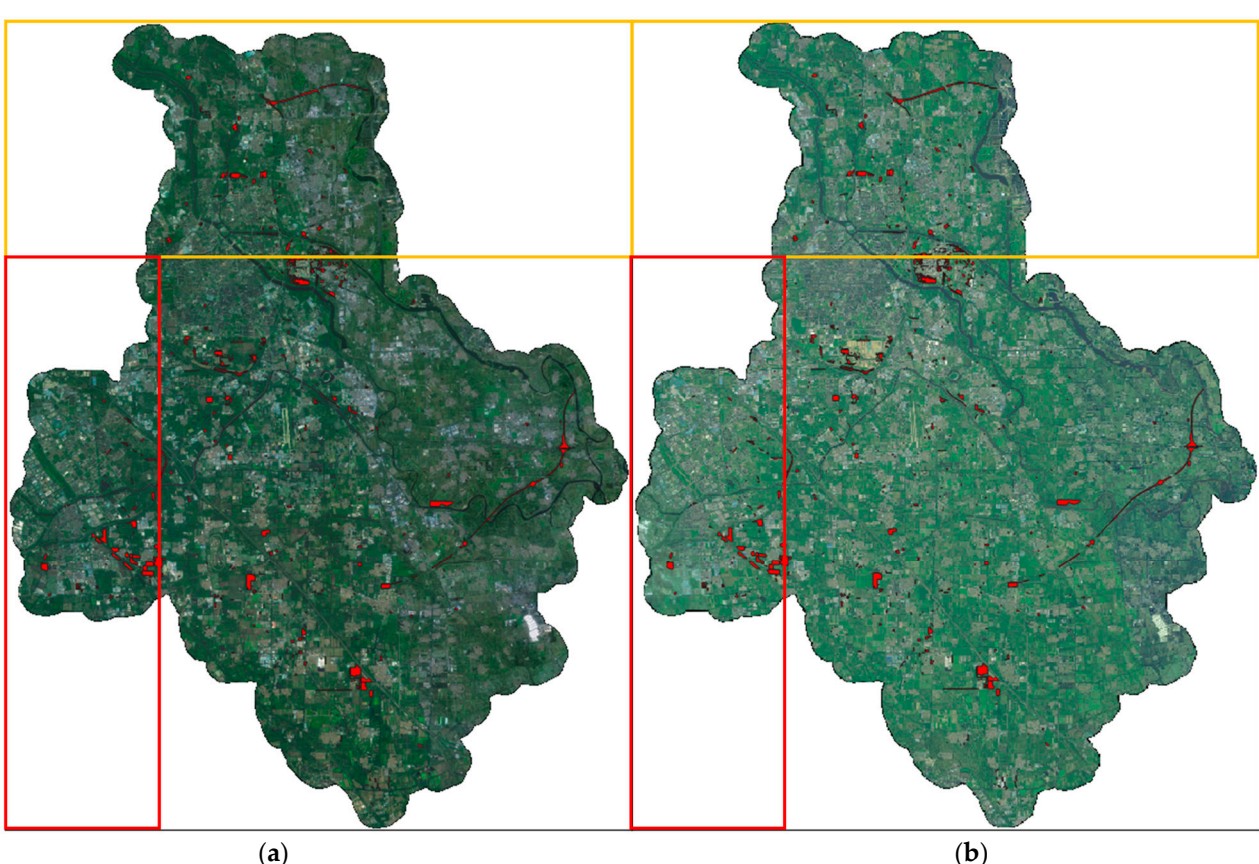

(**a**)                     (**b**)

**Figure 8.** The TZ-CD dataset. The red area represents the change label. The yellow box area is used for testing. The red box area is used for validation. The other area is used for training. (**a**) Post-image. (**b**) Pre-image.

As shown in Figure 9, some of the data from TZ-CD are plotted. As shown in the first column, a large area of vegetation changes to push-fill, while the second column contains four similar areas of change, three of which have more pronounced changes and the fourth has weaker changes. However, the changes in the other three areas can be inferred from their sending changes. The third column contains weaker road changes. The fourth column contains building changes. The fifth column contains the changes in smaller targets. The

examples show that the change scenarios are rich and cover a vast area compared to other publicly available datasets.

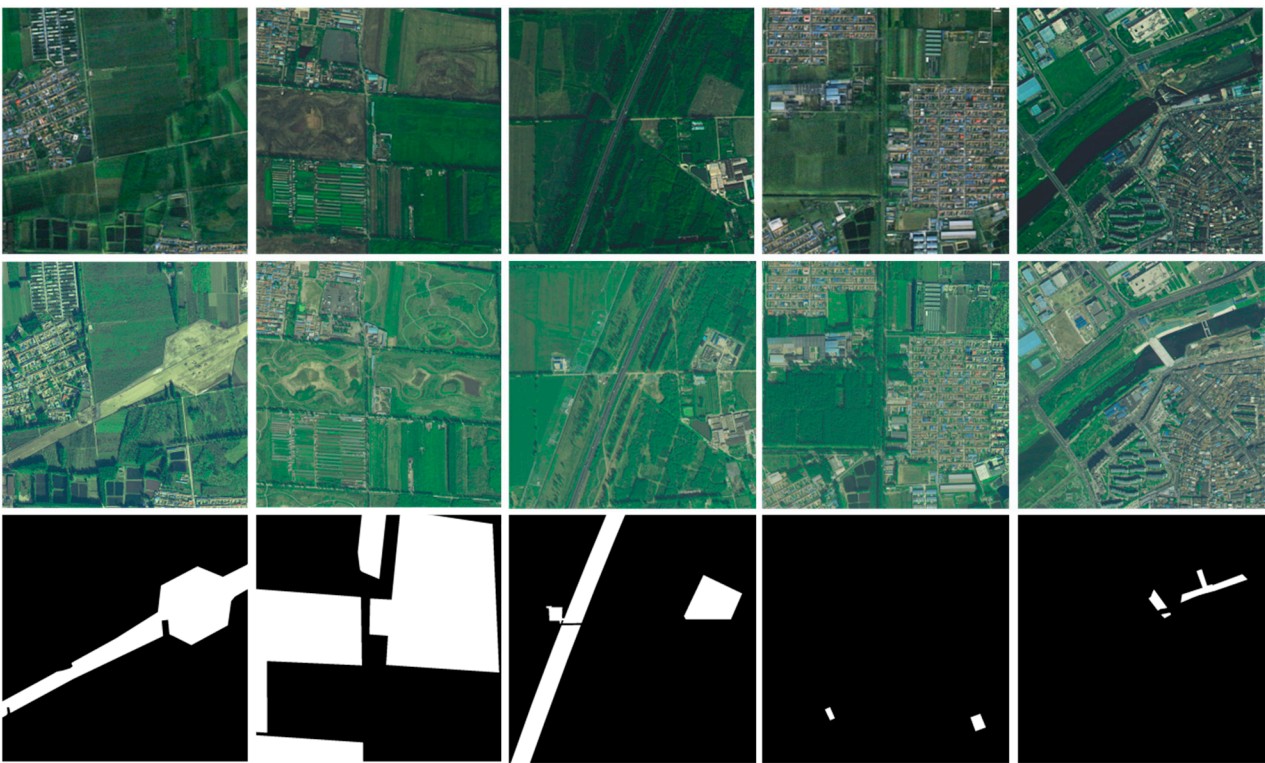

**Figure 9.** TZ-CD dataset examples, including the front post-image, pre-image and ground truth from top to bottom.

## 4. Experiments

### 4.1. Evaluation

LEVIR-CD, WHU-CD and TZ-CD are binary category datasets. We use the evaluation metrics precision, recall, F1-score and intersection over union (IOU), which are commonly employed for semantic segmentation. The values of precision, recall, F1-score and IoU range from 0 to 1. There are differences in the calculation of different assessment metrics, and multiple dimensions of metrics reflect the differences in the testing results in a more scientific way. The metrics are calculated as follows:

$$IoU = \frac{TP}{TP + FP + FN} \tag{16}$$

$$F1 = \frac{2 * TP}{2 * TP + FP + FN} \tag{17}$$

$$precision = \frac{TP}{TP + FP} \tag{18}$$

$$recall = \frac{TP}{TP + FN} \tag{19}$$

TP denotes true positives, *FP* denotes false positives, and *FN* denotes false negatives.

### 4.2. Implementation Details

In data preprocessing, LEVIR-CD provides a cropped completed dataset with a size of 1024 × 1024. For WUH-CD and TZ-CD, the data are cropped to 1024 × 1024 with a 0.1 overlap rate. The data are normalized from 0–255 to 0–1 during training and prediction.

In the training phase, the Adam [48] optimizer is selected with a ploy learning rate, 300 epochs, starting learning rate of 0.001, and ending learning rate of 0. Data enhancement includes flipping, scaling (0.8–1.2), cropping, rotation, HSV change, RGB enhancement, color jitter, and Gaussian noise.

In the prediction stage, LEVIR-CD uses $1024 \times 1024$ images for prediction and evaluation. WUH-CD and TZ-CD use large images for prediction and accuracy evaluation. Large image prediction uses sliding windows, the sliding window size is 1024, and the overlap rate is 0.1. To prevent edge semantic missing data, each sliding frame is filled with 256 images, which are cropped to a 1536 * 1536 area, and the prediction result is the middle $1024 \times 1024$ area.

TChange is based on the ptyroch [49] framework and runs on cuda11.1, with all experiments completed on a single RTX 3090.

### 4.3. Ablation Studies

To demonstrate the contributions of Change MSA, the ISTM and deep edge supervision, we carry out ablation studies on the TZ-CD dataset. The baseline models for comparison are TChange to remove the Change MSA, the ISTM and deep edge supervision proposed in this paper. All models use an encoder of efficientnet-b1 [50], and the training strategy is the same.

**Ablation Studies on Change MSA**: Change MSA provides global feature swapping and interchannel feature swapping, containing C-MSA, S-MSA, and FFM, whose ablation experiments are shown in Table 1. When C-MSA is used alone, its accuracy appears to decrease compared to the base, and when combined with the FFM, its IoU accuracy increases by 2.5% compared to the base, which can indicate that using the bias induction property of convolution to enhance the transformer's local feature modeling is necessary. When S-MSA is added for spatial global modeling, its IoU accuracy continues to grow by 1.2%.

**Table 1.** Ablation studies on Change MSA. Results on the TZ-CD dataset.

| Methods | Precision | Recall | F1 | IoU |
|---|---|---|---|---|
| base | 80.81 | 81.73 | 81.27 | 68.45 |
| Base + C-MSA | 75.74 | 82.76 | 79.10 | 65.42 |
| Base + C-MSA + FFM | 85.72 | 80.52 | 83.05 | 71.01 |
| Base + Change MSA | 84.79 | 83.05 | 83.91 | 72.27 |

Change MSA: C-MSA + FFM + S-MSA + FFM.

**Ablation Studies on the ISTM:** The ISTM provides interchannel communication, and since it needs to perform channel dimensional agreement between two different scales, the ablation experiments of the ISTM are performed based on Base + Change MSA. The results are shown in Table 2, which shows that the ISTM provides 1% IoU growth.

**Table 2.** Ablation studies on the ISTM. Results on the TZ-CD dataset.

| ISTM | Precision | Recall | F1 | IoU |
|---|---|---|---|---|
| No | 84.79 | 83.05 | 83.91 | 72.27 |
| Yes | 84.46 | 84.64 | 84.55 | 73.23 |

**Ablation Studies on Deep Edge Supervision**: Deep supervision is a common supervision scheme, and in the field of change detection, some researchers use pixel-level truth labels for deep supervision [22]. Unlike previous deep supervision, the supervised labels employed in this paper are edge labels instead of pixel-level change region labels because the transformer tends to extract low-frequency information and disregard high-frequency information, while the CNN tends to disregard high-frequency information. The ablation experiments also prove the effectiveness of this design, as shown in Table 3. Base is a pure CNN network, while TChange is a hybrid transformer–CNN network. It can be seen

that the Base network achieves higher performance when pixel-level labels are used for supervision than when edge labels are used for supervision.

**Table 3.** Ablation studies on deep edge supervision. Results on the TZ-CD dataset.

| Methods | Precision | Recall | F1 | IoU |
|---|---|---|---|---|
| Base | 80.81 | 81.73 | 81.27 | 68.45 |
| Base + deep supervision | 86.37 | 77.49 | 81.69 | 69.04 |
| Base + deep edge supervision | 81.24 | 81.97 | 81.60 | 68.92 |
| TChange | 84.46 | 84.64 | 84.55 | 73.23 |
| TChange + deep supervision | 91.18 | 80.00 | 85.21 | 74.23 |
| TChange + deep edge supervision | 90.42 | 81.11 | 85.51 | 74.70 |

### 4.4. Comparative Method

For the LEVIR-CD, WUH-CD and TZ-CD datasets, we compare our method with several state-of-the-art CD methods.

1.  FC-EF, FC-conc and FC-diff [15]: FC-EF follows the semantic segmentation scheme. First, the pre-image and post-image are overlapped. Second, the images are input to the encoder for encoding. Third, the segmentation result is obtained by the decoder. FC-conc replaces the single encoder in FC-EF with a twin encoder, the pairwise features are merged by a simple concatenation operation, and then the region change result is decoded by the decoder. FC-diff replaces the concatenate operator with diff for the pairwise feature fusion part based on FC-EF.
2.  BiDataNet [51]: BiDataNet is a single encoder change detection segmentation scheme that combines LSTM and UNet for better encoding of long-range information and communication of features between two different time phases.
3.  CDNet [52]: CDNet is a single encoder change detection network. This network is mainly utilized for road change detection, so its convolution size is modified from the commonly employed 3 * 3 to 7 * 7 to expand the receptive field.
4.  Unet++_MSOF [53]: Unet++_MSOF adds lateral connections to the regular Unet's short-cut connections to provide more communication channels with different scale features.
5.  DDCNN [18]: DDCNN proposes a dense attention method consisting of several upsampling attention units to model the internal correlation between high-level features and low-level features.
6.  BIT [40]: BIT is the first change detection network with a transformer. First, the convolutional neural network provides multiscale features. Second, 2 features are obtained by the transformer decoder. Last, the change region is obtained based on the difference in the features.
7.  DMATNet [41]: DMATNet is a dual-feature mixed attention-based transformer network. First, the twin encoder is used for feature extraction, Then, fuse the fine and coarse features with dual-feature mixed attention (DFMA) module.

### 4.5. Results on the LEVIR-CD

The experimental results of TChange in LEVIR-CD are shown in Table 4. TChange improves the f1 accuracy by nearly 3% compared to the transformer-based BIT model, and the intersection of the leading CNN models also improves the IoU accuracy by 2%. In Figure 10, the results of four sets of experiments are plotted. Each result contains the three leading models in LEVIR-CD, Unet++_MSOF, DDCNN, and TChange. The visualization results show that TChange has much lower error detection than Unet++_MSOF due to the enhanced relationship among the data by Change MSA. Additionally, the completeness of the results at the edges is significantly improved due to the deep supervision of the edges. UNet++_MSOF and DDCNN have higher precision but lower recall, which is more consistent with the reflection in the visualization that they have fewer missed detections and more false detections.

**Table 4.** Results on the LEVIR-CD dataset.

| Methods | Precision | Recall | F1 | IoU |
|---|---|---|---|---|
| FC-EF [15] | 79.20 | 81.72 | 80.44 | 67.27 |
| FC-conc [15] | 91.46 | 88.05 | 89.72 | 81.36 |
| FC-diff [15] | 89.92 | **89.00** | 89.45 | 80.92 |
| BiDataNet [51] | 92.50 | 87.14 | 89.74 | 81.39 |
| CDNet [52] | 90.73 | 86.82 | 88.73 | 79.75 |
| Unet++_MSOF [53] | **92.18** | 87.84 | **89.96** | **81.75** |
| DDCNN [18] | **91.85** | 88.69 | **90.24** | **82.21** |
| BIT [40] | 89.24 | **89.37** | 89.31 | 80.68 |
| DMATNet [41] | 91.56 | **89.98** | 90.75 | 84.13 |
| TChange | **93.47** | **91.94** | **92.27** | **85.65** |

Color description: **best**, **2nd best**, and **3rd best**.

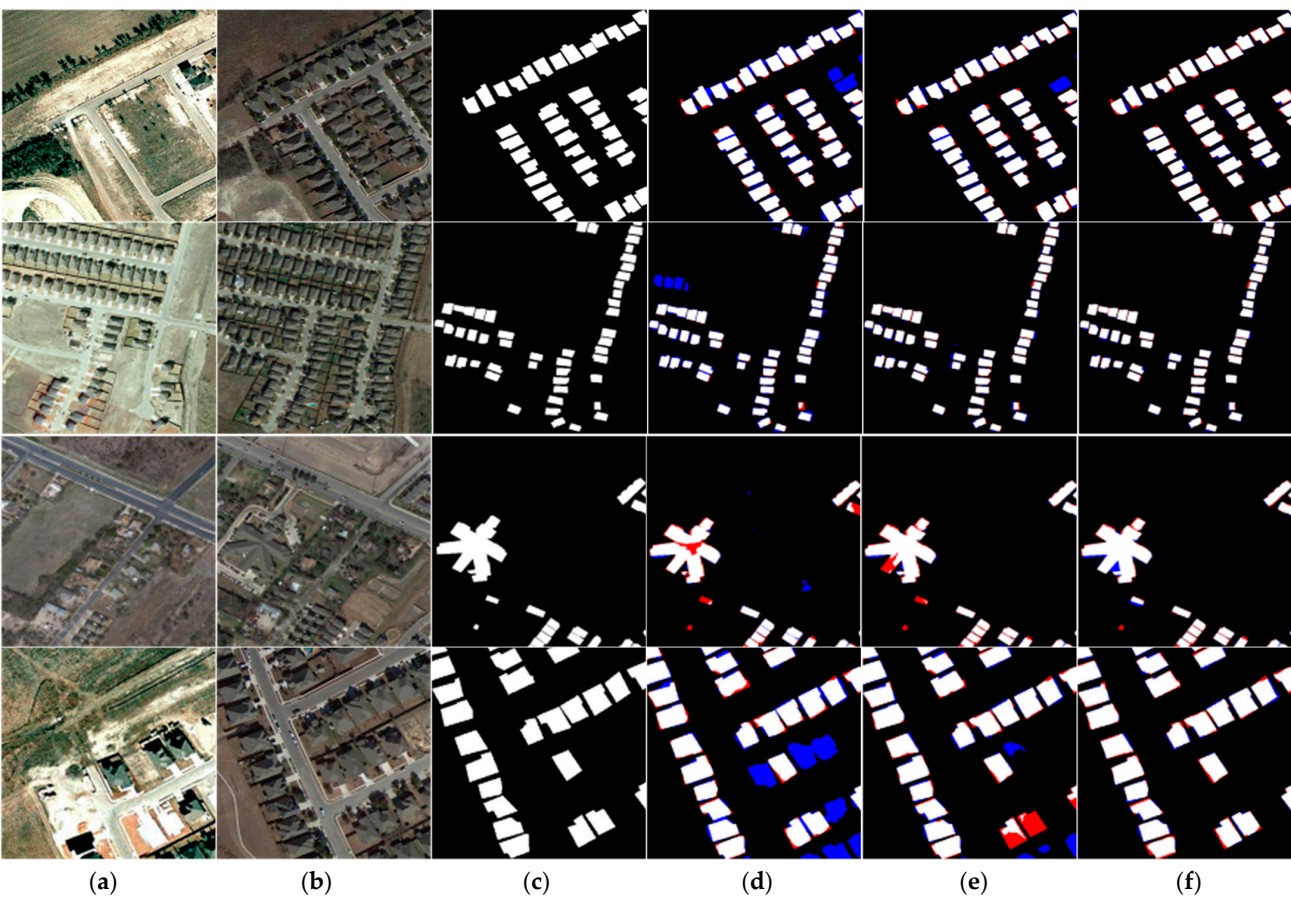

(a)      (b)      (c)      (d)      (e)      (f)

**Figure 10.** LEVIR-CD dataset results. The red area represents the missed area, and the blue area represents the wrong area. (**a**) Post-image. (**b**) Pre-image. (**c**) Ground truth. (**d**) Unet++_MSOF results. (**e**) DDCNN results. (**f**) TChange results.

### 4.6. Results on the WUH-CD

The experimental results of TChange in LEVIR-CD are shown in Table 5. With increasing resolution, the importance of multiple scales becomes increasingly obvious. Both BiDataNet and DDCNN construct more direct exchange channels for multiscale features, and their accuracy has more obvious advantages in the WUH-CD dataset where large targets are dominant. For the TChange proposed in this paper, due to stronger in-scale global feature modeling and the use of advanced transformer for interchannel information exchange, its F1 accuracy i improves 1.1% compared to the state-of-the-art algorithm. The visualization results in Figure 11 also show that TChange achieves complete detection of large targets and distinct edges.

**Table 5.** Results on the WUH-CD dataset.

| Methods | Precision | Recall | F1 | IoU |
|---|---|---|---|---|
| FC-EF [15] | 85.27 | 75.11 | 79.87 | 66.49 |
| FC-conc [15] | 92.67 | 55.59 | 69.49 | 53.25 |
| FC-diff [15] | 88.76 | 66.40 | 75.97 | 61.25 |
| BiDataNet [51] | **93.44** | **87.66** | **90.46** | **82.58** |
| CDNet [52] | 87.64 | 81.43 | 84.42 | 73.04 |
| Unet++_MSOF [53] | 93.03 | 86.29 | 89.53 | 81.05 |
| DDCNN [18] | 96.11 | 89.73 | 92.81 | 86.59 |
| BIT [40] | 86.64 | 81.48 | 83.98 | 72.39 |
| DMATNet [41] | 83.87 | 82.24 | 85.70 | 74.98 |
| TChange | 94.70 | 93.22 | 93.96 | 88.60 |

Color description: **best**, **2nd best**, and **3rd best.**

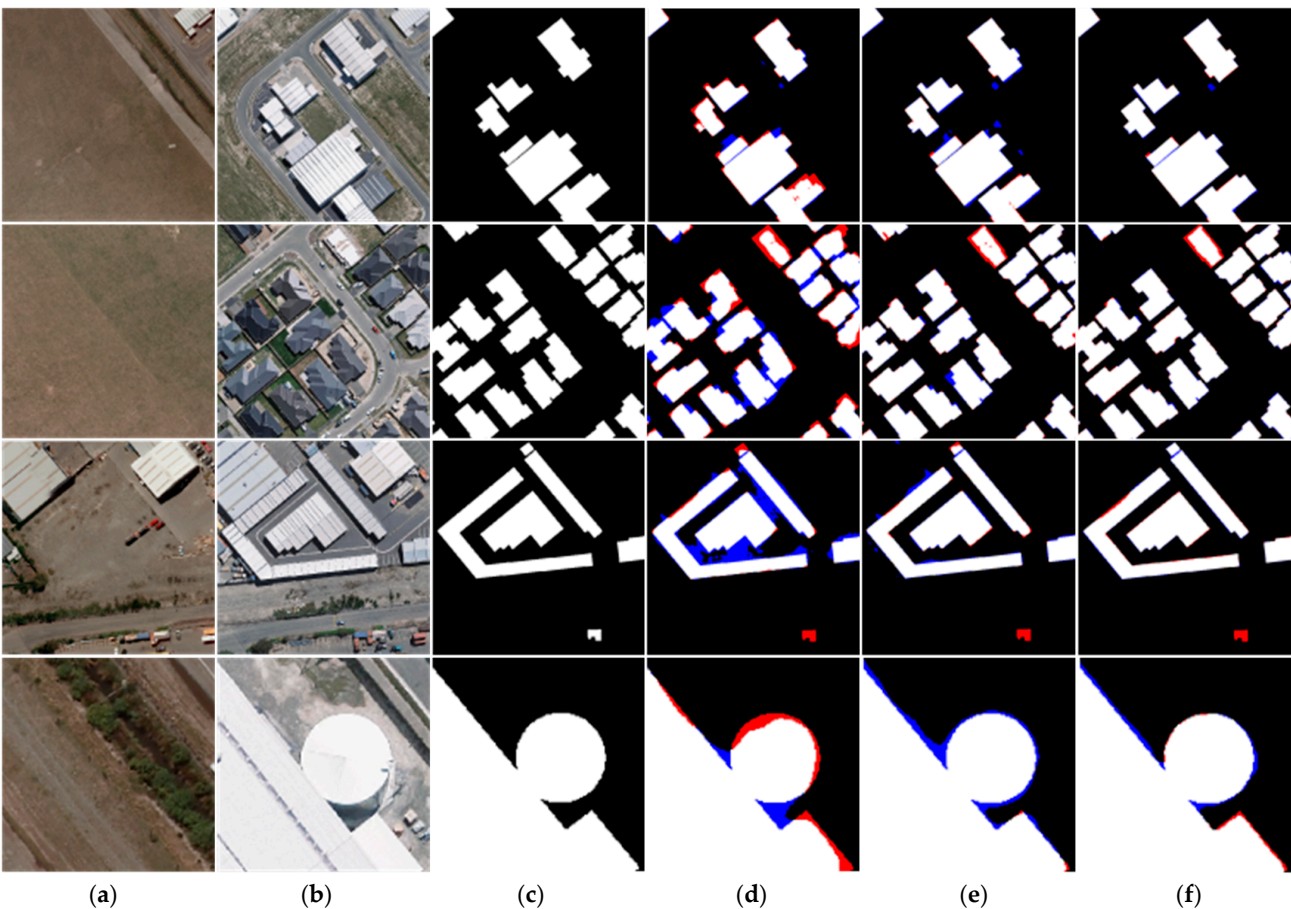

(a) (b) (c) (d) (e) (f)

**Figure 11.** WUH-CD dataset results. The red area represents the missed area, and the blue area represents the incorrect area. (**a**) Post-image. (**b**) Pre-image. (**c**) Ground truth. (**d**) BiDataNet results. (**e**) DDCNN results. (**f**) TChange results.

*4.7. Results on the TZ-CD*

The experimental results of TChange in TZ-CD are shown in Table 6. In the TZ-CD dataset presented in this paper, the proposed TChange model has an advantage over other models due to the many large targets included, and its F1 accuracy is 3% greater than the current leading models. In Figure 12, the results of TChange on the full map of the TZ-CD test set are shown. Three areas are intercepted in the full map for display. In the first set of predicted data, a slender change is completely detected, while in the second set, there is a pooling area, resulting in a weak change. TChange is able to completely identify and extract the change area. In the third set, due to the combination of the transformer and CNN, TChange is able to perform a more complete extraction for oversized targets.

**Table 6.** Results on the TZ-CD dataset.

| Methods | Precision | Recall | F1 | IoU |
|---|---|---|---|---|
| FC-EF [15] | 47.66 | **82.29** | 60.36 | 43.23 |
| FC-conc [15] | 78.79 | 74.16 | 76.41 | 61.82 |
| FC-diff [15] | 57.78 | **90.81** | 70.62 | 54.59 |
| BiDataNet [51] | 67.93 | **86.40** | 76.06 | 61.37 |
| CDNet [52] | 73.62 | 75.69 | 74.64 | 59.55 |
| Unet++_MSOF [53] | **79.90** | 80.11 | **80.00** | **66.68** |
| DDCNN [18] | **89.00** | 76.90 | **82.50** | **70.21** |
| BIT [40] | 73.58 | 82.04 | 77.58 | 63.37 |
| TChange | **90.42** | 81.11 | **85.51** | **74.70** |

Color description: **best**, **2nd best**, and **3rd best**.

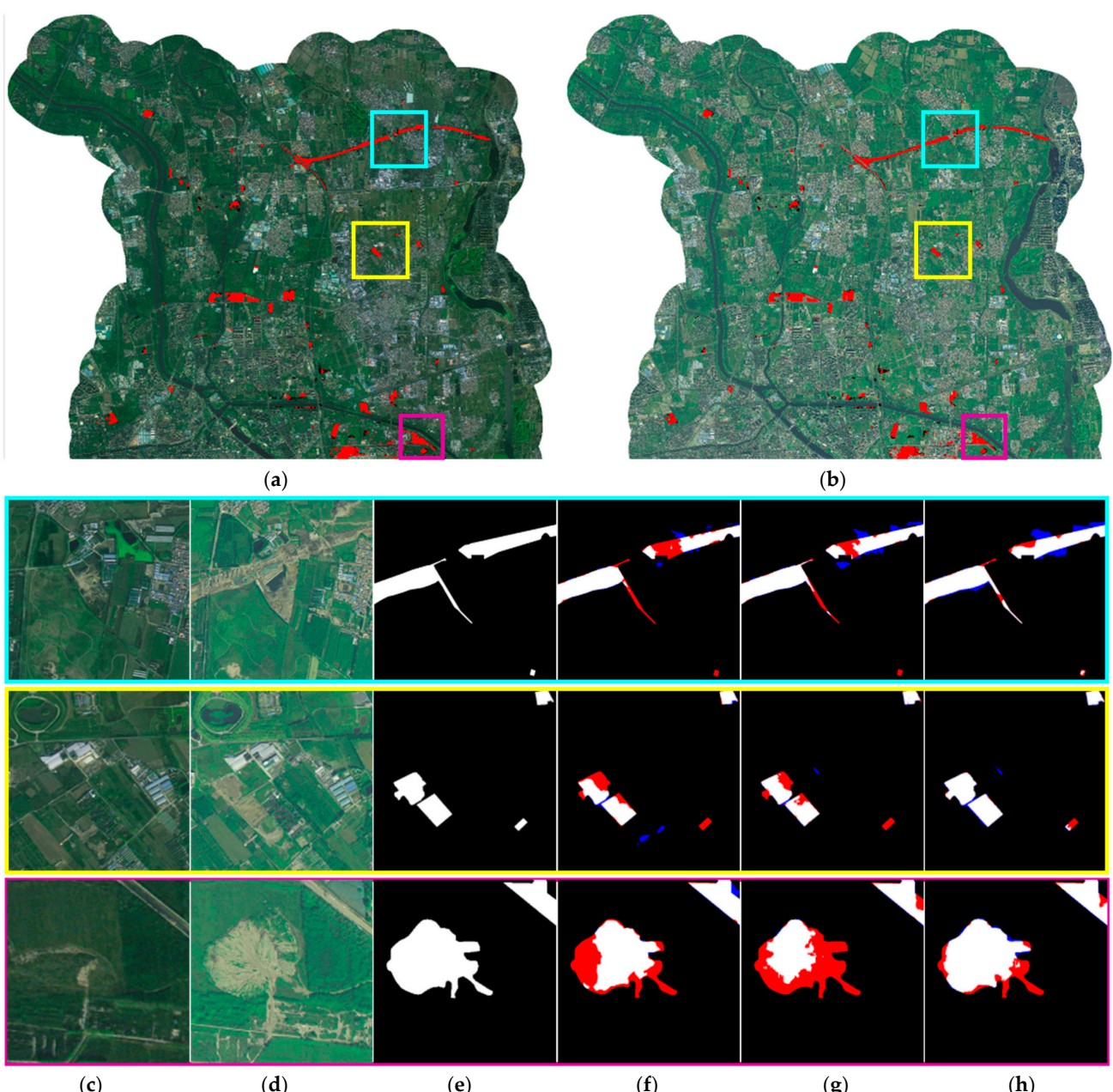

**Figure 12.** WUH-CD dataset results. The red area represents the missed area, and the blue area represents the incorrect area. (**a**) Pre-large image; (**b**) pre-large image, three areas from the large image, represented by three colors; and (**c**) postimage. (**d**) Pre-image. (**e**) Ground truth. (**f**) BiDataNet results. (**g**) DDCNN results. (**h**) TChange results.

*4.8. Efficiency Comparison Experiments*

There is a constraint relationship between the accuracy efficiency of the deep learning model, so in this study, based on the WUH-CD dataset, efficiency comparison experiments are conducted using a single 3090 graphics card, and the results are shown in Table 7. The accuracy achieved by the network proposed in this paper is obviously superior at the same scale.

**Table 7.** Efficiency comparison in the WHU-CD dataset.

| Methods | Params (Mb) | Inference Time (s) |
| :---: | :---: | :---: |
| FC-EF [15] | 5.2 | 21 |
| FC-conc [15] | 5.9 | 21 |
| FC-diff [15] | 5.2 | 21 |
| BiDataNet [51] | 7.8 | 26 |
| CDNet [52] | 34.3 | 27 |
| Unet++_MSOF [53] | 178.2 | 44 |
| DDCNN [18] | 121.2 | 46 |
| BIT [40] | 132.9 | 63 |
| TChange | 166.2 | 70 |

## 5. Conclusions

In this paper, we propose a hybrid transformer–CNN network (TChange) for the high-resolution change detection task, which can efficiently extract multiscale features by using lightweight efficient b1 in the encoder part. For the problem that large targets are prone to voids, Change MSA is proposed to acquire long-range information in both the channel dimension and spatial dimension. To address the problem of easily missed detection for both large targets and small targets, we propose IMST based on the transformer to construct direct communication channels between two scales. In the CNN decoder, the features from the transformer are simultaneously input, and the multiscale features extracted by the encoder are fused to obtain the final change region. For the problem that the transformer tends to lose high-frequency features, the use of deep edge supervision is proposed to replace the commonly employed depth supervision.

To verify the effectiveness of the proposed algorithm, a large-scale change building dataset, TZ-CD, is constructed. The dataset contains both very large targets and weak feature targets, which are lacking in the public dataset, so that the performance of the algorithm can be more comprehensively measured. TChange achieves state-of-the-art results in both the two open-source datasets and TZ-CD.

Although TChange has improved its accuracy compared with the previous network and performed well in the public dataset, the current change detection algorithm, as exposed by the TZ-CD dataset, still cannot achieve complete extraction when the task is more complex. Moreover, the dataset used in this paper has basically the same distribution between the training set and the test set, and the experimental environment is more ideal. It can be expected that when domain differences appear, the accuracy differences between two different models will be further widened, and more problems will be exposed. Therefore, in subsequent work, we will test the performance of the algorithm for complex tasks and domain differences and then identify and solve any problems.

**Author Contributions:** Conceptualization, Y.D. and Y.M.; methodology, Y.D., Y.M. and J.C. (Jingbo Chen); validation, Y.D., D.L. and A.Y.; data curation, J.C. (Jing Chen); writing—original draft preparation, Y.D.; writing—review and editing, Y.M. and J.C. (Jingbo Chen); All authors have read and agreed to the published version of the manuscript.

**Funding:** This research was funded by the National Key R&D Program of China (under grant number: 2021YFB3900503) and the National Natural Science Foundation of China (under grant number: 61901471).

**Data Availability Statement:** Not applicable.

**Conflicts of Interest:** The authors declare no conflict of interest.

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
