# Peer review of "TChange: A Hybrid Transformer-CNN Change Detection Network"

_remotesensing, doi:10.3390/rs15051219_

Round 1
Reviewer 1 Report
This paper is well written and there are still some comments need to be revised before acceptance for publication.
(1) The reason for designing such a structure of the deep learning model shown in Figure 1 should be introduced in detail.
(2) Since the model contains multiple modules, how can its convergence be guaranteed. Perhaps the convergence curve of the objective function of the model should be shown in the paper.
(3) The computational complexity of the model should be analyzed, or the reasoning speed should be reported.
(4) The generalization and regularization of deep learning models across various scenes and conditions has not been discussed in the Introduction or Related Work.The following related work should be cited and analyzed in Section 1, including “Faster mean-shift: GPU-accelerated clustering for cosine embedding-based cell segmentation and tracking.” Medical Image Analysis 71 (2021): 102048. “Compound figure separation of biomedical images with side loss,” Deep Generative Models, and Data Augmentation, Labelling, and Imperfections. Springer, Cham, 2021. 173-183. “Pseudo RGB-D Face Recognition,” in IEEE Sensors Journal, vol. 22, no. 22, pp. 21780-21794, 15 Nov.15, 2022. “Improvement of generalization ability of deep CNN via implicit regularization in two-stage training process,” IEEE Access, vol. 6, pp. 15844-15869, 2018.
Reviewer 2 Report
Line 18: "MSA" is not defined until line 100.
The manuscript proposes a deep learning approach for detection of changes in high-resolution images. Image generation techniques based on self-attention allowed for extraction of features that are significantly smaller than in previously reported work. The authors develop further the long-existing change detection research and although improvement in results are just few percent, these improvements may be important in certain areas such as military applications. The paper also introduces a new dataset used in experiments that is the largest dataset of a kind so far. The results of experiments are convincing and present an improvement over the previously reported results by other researchers. The conclusions are consistent with the claims and included examples substantiate these claims. The references are appropriate, sufficiently recent and cover the discussed subject. Figure captions are sufficiently extensive. The manuscript is suitable for publication after the improvement of language.
Line 45: "change regions" - better "changed regions" or "regions of change"
Line 48: "as far" means "as different" or "as close"?
Line 81: what does "directly concate" means? Concatenate?
Line 197: "multi-head"
Line 247: "SWIN" (Sliding window?) is not defined.
Line 320: "concatenated"
Line 329: What is "se attention"?
Line 408: "training"?
Is there a reason why four scales are considered for ChangeMSA (Fig.1)? Later, Line 173, there are five scales.
"IoU", "IOU", "iou" - are these the same, or have different meanings?
Reviewer 3 Report
General Comment:
This manuscript presents a novel supervised change detection method based on Transformer and CNN models, aiming to overcome two chllenges, which are concretized as the intra-scale and inter-scale problems. The authors proposed a new MSA module, a new feature fusion method, and an inter-scale transformer module, then integrated them and introduced a Transformer-CNN network to detect (building) changes in bi-temporal remote sensing images. From my perspective, the manuscript is well-motivated, structed, and written. Nevertheless, the experimental part of this paper still needs some supplementations and improvements.
Specific Comments:
1. The title of this research seems to be of ambiguity. The phrase 'hybrid change detection' has not previously appeared in other relevant scientific papers, what does it denote? Placing 'hybrid' before 'Transformer-CNN' or deleting 'hybrid' may be better.
2. A comprehensive review of the recent researches of adopting transformer networks to promote the performance of change detection models using remote sensing imagery would be beneficial to this manuscript.
3. Please correctly cite the paper 'Swin transformer: Hierarchical vision transformer using shifted Windows' in the text. Insert the reference number when referring to this paper rather than using SWIN instead.
4. The authors mentioned that 'original MSA ignores the biased induction characteristics of CNN' (in line 96 to line 97) and the authors prposed the FFM to 'use offset induction of convolutional net- 257 work to obtain local features, which is used to strengthen the global feature modeling of 258 MSA' (in section 2.3.3). The FFM is just a convolutional layer, it seems to me that a lot more improvements can be taken rather than merely use one convolutional layer for its inductive bias of locality if the authors aim to enhance the local feature learning.
5. The abbreviations of concepts should be explained when they first existed in the text.
6. Reference 36 and Reference 43 are the same paper, please carefully check the Reference part.
7. If this constructed dataset is one of the contributions of this research, as described in Section I. Introduction, it needs to be uploaded to the official database or Google drive for utilization and competition by other scholars.
8. Among all the baseline competitors, there is only one transformer-based method. It is suggested to add more latest transformer-based methods into the comparation, as shown below.
1) Zhang, C.; Wang, L.; Cheng, S.; Li, Y. SwinSUNet: Pure Transformer Network for Remote Sensing Image Change Detection. IEEE TGRS 2022.
2) Song, X., Hua, Z., Li, J. Remote Sensing Image Change Detection Transformer Network Based on Dual-Feature Mixed Attention. IEEE TGRS 2022.
Reviewer 4 Report
The authors propose a transformer-CNN hybrid method for change detection task. The method employs a Siamese encoder, a hybrid transformer-CNN decoder to achieve SOTA performance. There are some unclear parts that need further explanation. Some specific comments are provided below.
1. 2.3.2 concisely describes S-MSA, which seems to be similar to the block MAS of SWIN. Please describe it if possible
2. In Change MSA, C-MSA and S-MSA are connected by serial design. Can we connect or exchange the current S-MSA and C-MAS sequences in parallel? Please give further instructions or experiments.
3. Some formats of references need to be checked. For example, there are no journal names for Ref 15, Ref.16. Ref46, and et al
4. Some grammatical errors of the title of figures should be revised, such as Figure 4, Figure 5 and et al.
5. To better motivate this paper, the authors are suggested to discuss and review more recently published methods, such as:
(1) A Change Detection Method Based on Multi-Scale Adaptive Convolution Kernel Network and Multimodal Conditional Random Field for Multi-Temporal Multispectral Images, Remote Sensing, 2022
(2) D-former: A u-shaped dilated transformer for 3d medical image segmentation, Neural Computing and Applications, 2022.
(3) Spatial–spectral attention network guided with change magnitude image for land cover change detection using remote sensing images, IEEE Transactions on Geoscience and Remote Sensing, 2022.
Round 2
Reviewer 1 Report
Accept.
Author Response
Thank you for your recognition and support of my work
Reviewer 3 Report
I have no further comments except for three minor issues as follows.
1. There are two Tables are entitled 'Table 6', in Page 17 and Page 19, respectively.
2. In the second Table 6, why are some precision values larger than 100 or 100%?
3. Why are there numerous signs '%@' in Section References?
Author Response
Thank you very much for the problem you pointed out. Due to my negligence, I uploaded the wrong version. We have changed table6 to table7, and table7 is based on the experimental results of model size and efficiency, so there will be a situation of more than 100. Finally, in references to, correct the error of '%@', thank you again for your tolerance, best wishes.

Reviewer 4 Report
No more comments
Author Response

(The authors gave the same response as above.)
